# Geospatial analysis of leptospirosis clusters and risk factors in two provinces of the Dominican Republic

Beatris Mario Martin[1]*, Benn Sartorius[1], Helen J. Mayfield[1], Angela M. Cadavid Restrepo[2], Behzad Kiani[1], Cecilia J. Then Paulino[3], Marie Caroline Etienne[4], Ronald Skewes-Ramm[3], Michael de St. Aubin[4,5], Devan Dumas[4,5], Salomé Garnier[4,5], William Duke[6], Farah Peña[3], Gabriela Abdalla[4], Lucia de la Cruz[3], Bernarda Henríquez[3], Margaret Baldwin[4,5], Adam Kucharski[7], Eric J. Nilles[4,5,8], Colleen L. Lau[1]

**1** Centre for Clinical Research, Faculty of Health, Medicine and Behavioural Sciences, The University of Queensland, Brisbane, Australia, **2** School of Public Health, Faculty of Health, Medicine and Behavioural Sciences, The University of Queensland, Brisbane, Australia, **3** Ministry of Health, Santo Domingo, Dominican Republic, **4** Brigham and Women's Hospital, Boston, Massachusetts, United States of America, **5** Harvard Humanitarian Initiative, Cambridge, Massachusetts, United States of America, **6** Pedro Henríquez Ureña National University, Santo Domingo, Dominican Republic, **7** London School of Hygiene & Tropical Medicine, London, United Kingdom, **8** Harvard Medical School, Boston, Massachusetts, United States of America

* beatris.martin@uq.edu.au

## Abstract

### Background

Drivers of leptospirosis transmission can vary across regions, leading to spatial clustering of infections. This study aims to identify clusters of leptospirosis seroprevalence in the Dominican Republic (DR) and factors associated with high-risk areas.

### Methodology/Principal Findings

We analysed data from two provinces, Espaillat and San Pedro de Macoris (SPM), obtained on a national survey conducted in 2021 (n = 2,078). Samples were tested by microscopic agglutination testing (MAT) to detect leptospirosis antibodies. We used flexible spatial scan statistics to locate significant clusters for seropositive individuals (all serogroups combined) in each province and calculated risk ratios (RR) at the household and community level. Environmental and sociodemographic risk factors associated with clusters were assessed by logistic regression. One cluster was identified in each province. Participants living inside a cluster were more likely to live further from health facilities (OR 1.86, p < 0.001 and OR 4.41, p = 0.044 by motorized travel time in Espaillat and SPM, respectively). Cluster participants were also less likely to live in areas of higher population density (OR 0.76, p < 0.01 and OR 0.29, p < 0.001 in Espaillat and SPM, respectively) and in communities with higher gross

**Data availability statement:** A de-identified dataset is available at https://github.com/enilles1/DR-Leptospirosis for the purpose of reproducing and building on the analyses. The cluster analysis was conducted using SaTScan.

**Funding:** This research was funded by the US CDC U01, grant number U01GH002238. CLL was supported by Australian National Health and Medical Research Council Fellowships (1109035 and 1193826). BS, HJM and BK were supported by The University of Queensland's Health Research Accelerator (HERA) initiative (2021-2028). CDC staff supported laboratory analysis. Funders had no role in study design, data collection and analysis, decision to publish, or preparation of the manuscript.

**Competing interests:** The authors have declared that no competing interests exist.

domestic product (GDP) (OR 0.70, p < 0.001 and OR 0.42, p < 0.001 in Espaillat and SPM, respectively). Additional risk factors varied between Espaillat and SPM.

## Conclusion/Significance

Our findings confirm the clustered spatial pattern of leptospirosis and highlight that transmission drivers vary by province. While both provinces show higher transmission in impoverished areas, modifiable factors differ, requiring tailored public health interventions.

## Author summary

Leptospirosis is a significant public health problem in Latin America and the Caribbean, accounting for one-third of all reported outbreaks globally between 1970 and 2012. In the Dominican Republic (DR), 2,860 human leptospirosis cases were reported to the General Epidemiology Directorate of the Ministry of Public Health and Social Assistance between 2013 and 2023. The country's warm and humid climate facilitates leptospirosis transmission. However, environmental and sociodemographic drivers can vary across regions, resulting in a geographically heterogeneous distribution of infection. This study aimed to identify areas with a higher prevalence of leptospirosis seropositivity (clusters) of leptospirosis seroprevalence in two provinces of the DR using flexible spatial scan statistics. This approach allowed us to investigate the existence of clusters in each province. Additionally, we used logistic regression to identify environmental and sociodemographic drivers associated with clusters in each province. Our findings suggest the presence of clusters in both provinces, with different sets of significant drivers identified in each province. Notably, drivers associated with clusters in both provinces were highly indicative of socioeconomically vulnerable populations, highlighting leptospirosis as a disease of poverty. These results underscore the need for geographically targeted and tailored interventions to reduce leptospirosis disease burden in at-risk communities.

## Introduction

Leptospirosis is one of the most widespread zoonotic disease globally [1]. The *Leptospira* bacteria present a complex transmission cycle that can include several species of mammalian reservoirs. These reservoirs shed pathogenic serovars through their urine, contaminating water and soil where the bacteria can survive for long periods [2]. The disease is considered to be primarily an occupational infection, associated with groups who handle animals or animal tissues and subsistence farming in tropical countries. In these regions, leptospirosis exhibits an endemic pattern of occurrence mainly in rural settings [3]. It is also considered to be an emerging infectious disease with outbreaks frequently associated with extreme rainfall events, such as hurricanes and flooding [2,4,5], and often cited as an urban slum health problem in resource-poor countries [6,7].

Annually, leptospirosis causes an estimated one million cases and around 60,000 deaths worldwide [8], though the actual burden may be higher due to underreporting. The reference method for serological diagnosis is the microscopic agglutination test (MAT), which can be time-consuming, and availability is frequently limited, especially in low and middle-income countries [1]. Tropical islands are considered a particular high-risk setting for human infection [8,9] and a recent study finding one-third of all globally reported outbreaks occurred in Latin America and the Caribbean (LAC) [10]. While recent outbreaks of the disease have been primarily associated with floods and hurricanes, this pattern coexists with the endemic form in this region. Drivers and risk factors for each epidemiological context are specific and remain poorly explored in most countries [11].

The adoption of spatial modelling methods for studying infectious diseases has contributed to a better understanding of the distribution of seroprevalence and risk factors and drivers of many diseases [12–14], including leptospirosis [15]. Cluster analysis methods are spatial statistical tests that assess geographical variation in disease risk and/or occurrence, and identify areas where the number of events (e.g., participants testing positive for leptospirosis seromarkers) exceeds expectations based on the size of the population analysed [16]. Increasing evidence suggests a spatial clustering of leptospirosis infection [17–21], indicating that some areas may provide more favourable conditions for sustaining transmission cycles than others.

The complex interactions between ecology, epidemiology, multiple animal hosts, and human conditions in urban and rural settings, identifying characteristics associated with populations living in high-risk areas can provide insights to guide targeted public health action. However, the importance of risk factors and drivers can vary across space [22] due to local differences in the complex transmission cycle [23], such as the presence of different reservoir mammals and their intrinsic relationship with serovars [5], vaccination rate of livestock herds [24], suitable conditions for pathogen survival in the environment [2] and opportunities for human exposure [1,5]. The variation of suitable conditions for transmission can occur within and between regions, with high-risk areas experiencing more localised outbreaks [17,20,21]. By pinpointing high-risk areas, spatial cluster analysis enables healthcare managers to generate hypotheses for understanding higher prevalence in these areas. This study aims to identify clusters of leptospirosis seroprevalence in the Dominican Republic (DR) and determine the factors associated with these high-risk areas.

## Methods

### Ethics Statement and consent

This study was approved by the National Council of Bioethics in Health (013–2019), the Institutional Review Board of Pedro Henríquez Ureña National University, Santo Domingo, DR; the Mass General Brigham Human Research Committee, Boston, USA (2019P000094); and the Human Research Ethics Committee of The University of Queensland (2022/HE001475), Brisbane, Australia.

Written consent was obtained from all participants. For participants <18 years old, except emancipated minors, consent was obtained from the parent or legal guardian. Participants between 14–17 years old provided written assent and those between 7–13 years old provided verbal assent. For participants between 5–7, written only parental consent was obtained.

### Study area

The Dominican Republic is located in the Caribbean region, sharing the island of Hispaniola with Haiti. It is the second most populous country in the region, after Cuba [25]. The country is divided into 31 provinces plus the Santo Domingo National District. These provinces and the National District are aggregated into 10 administrative regions: Cibao Norte, Cibao Sur, Cibao Nordeste, Cibao Noroeste, Valdesia, Enriquillo, El Valle, Yuma, Higuamo, Ozama o Metropolitana [26].

For the analysis presented here, data from two provinces were included, Espaillat – located in Cibao Norte region – and San Pedro de Macoris – in Higuamo region. These two provinces were selected because they were included in a

linked study on clinical surveillance of acute febrile illnesses (S1 Text). Espaillat and SPM are home to approximately 230,000, and 290,000 people, respectively. Both provinces have a predominantly young population (mean age 30.3 in Espaillat and 27.9 in SPM), evenly distributed between males and females [27]. However, there are notable sociodemographic and economic differences between the two provinces. In Espaillat, 54.7% of the population reside in rural setting, compared to 15.9% in SPM [27]. Economical activities developed in this setting also vary between these two provinces, in Espaillat, most agricultural units are focused on animal husbandry, whereas in SPM, they engage in a mix of animal husbandry and crop production [28].

### Data acquisition and processing

**Survey.** A three-stage cross-sectional national serosurvey was conducted in the DR, between 30 June and 12 October 2021. The survey sampling design has already been described previously [29]. In summary, a total of 10 communities were sampled from Espaillat Province and 13 from SPM. In both Espaillat and SPM, 60 households were selected from each community. All household members above 5 years of age were invited to participate. Participants included in the study were interviewed by a trained local fieldwork team. The questionnaire included individual and household-level questions. Venous blood was collected from all participants, and household Global Positioning System (GPS) coordinates were recorded.

Venous blood was processed as sera, and frozen at -80°C. Samples of participants from the two provinces were tested by MAT to detect anti-*Leptospira* antibodies. Serological analyses were performed at the Centers for Disease Control and Prevention's Zoonoses and Select Agent Laboratory, Bacterial Special Pathogens Branch, Atlanta, USA. A panel of 20 pathogenic serovars were selected for the MAT panel, and titres ≥1:100 were considered seropositive and indicative of prior infection. Serovars identified by MAT were used to determine the putative serogroups associated with infections, as previously described [30]. In summary, analyses were conducted on the serogroup level. If a sample reacted to multiple serogroups, the serogroup with the highest titre was considered the main one. When the highest titre was observed across two or more serogroups the sample was recorded as 'mixed'.

**Spatial data.** Environmental, sociodemographic and census data was obtained from publicly available sources. All spatial data considered for the analyses are described in Table 1. These data were extracted for the household location of each participant and incorporated into the survey data.

### Statistical analyses

**Global and local spatial cluster analysis.** Spatial autocorrelation refers to the phenomenon where events located near each other tend to be more similar than those further apart [39]. Global cluster analysis is applied to determine if spatial autocorrelation is present in the data, in other words if the occurrence of events is clustered across space. Local cluster analysis is used to identify and locate the clusters of events [40]. In this study, both types of clustering analyses were performed separately for each province, for all serogroups combined as well as the main serogroups identified.

To assess global spatial clustering, we used semivariograms, which are graphical representations of how the semivariance (y-axis) changes based on distance between pairs of observations (x-axis) [41,42]. If events are spatially autocorrelated, as the distance increases, so does the semivariance. The semivariogram is characterized by three main parameters: the *sill*, the *nugget* and the *range*. The *sill* is the value where the semivariance levels off, indicating that beyond this point, the spatial relationship weakens. The *nugget* is the value where the curve meets the y-axis. It represents the measurement error or very short-range spatial variability. The *range* is the distance at which the semivariance reaches the *sill*. It shows the maximum distance over which autocorrelation exists. If the curve does not plateau, this suggests that there is no significant spatial autocorrelation (S1 Fig. To identify the proportion of variation due to the spatial structure, the difference between the *sill* and the *nugget* is divided by the sum of the *sill* and the *nugget*. A value below 0.25 indicates a strong spatial autocorrelation.

**Neglected Tropical Diseases**

**Table 1. Environmental and sociodemographic data considered for the characterisation of high-risk areas of leptospirosis seroprevalence in the Dominican Republic, Jun-Oct, 2021.**

| Data | Description | Reference |
|---|---|---|
| Rural-urban classification | A map of the rural and urban barrio/paraje administrative boundaries was downloaded from the Dominican Republic National Statistics Office. | [31] |
| Distance to major roads | A raster with the distance (in km) from the cell centre to the nearest major road was downloaded from the WorldPop website at a resolution of 3 arc-seconds. | [32] |
| Distance to the provincial capital city | The location of provincial capitals in the DR was extracted in a shapefile format from the Dominican Republic National Statistics Office. A raster layer of the Euclidean distance between each household location and the nearest provincial capital was generated in km. | [31,33] |
| Distance to education facilities | Geographic locations of education facilities, including a mix of public and private institutions (kindergartens, schools, colleges and universities), were acquired from OpenStreetMap in a shapefile format. The Euclidean distance between each household and the nearest education facility was derived in km. | [33,34] |
| Walking travel time to health-care facilities | Estimates of travel time (in minutes) from each household to the nearest geolocated hospital or clinic were downloaded from the Malaria Atlas Project website. For locations where there was no data, a 1km buffer around the household was created, and the mean value on valid data points was extracted and assigned to the household. | [35] |
| Motorized travel time to healthcare facilities | Estimates of travel time (in minutes) from each household to the nearest hospital or clinic by motorized transport were extracted in from the Malaria Atlas Project website. For locations where there was no data, a 1km buffer around the household was created, and the mean value on valid data points was extracted and assigned to the household. | [35] |
| Elevation | Data were obtained at a resolution of 3 arc-second from the Shuttle Radar Topographic Mission (SRTM) dataset: SRTM 2000, Dominican Republic. For elevation the value of each grid cell represents its elevation above sea level in metres (m). For locations where there was no data, a 1km buffer around the household was created, and the mean value on valid data points was extracted and assigned to the household. | [36] |
| Precipitation | Monthly gridded rainfall time series from 2017 to 2021 was downloaded from CHIIRPS: rainfall estimates from rain gauge and satellite observations. This data set combines a 0.05° resolution satellite imagery and in-situ station data to create the rainfall grid. Data is provided in mm per month. Data extracted was aggregated to create a monthly average, maximum and minimum for the 5-year and, and 5-year average, maximum and minimum. | [37] |
| Population density | Estimates of population density for 2020 were downloaded from the WorldPop website. A raster was available for the DR at the resolution of 30 arc-seconds (approximately 1km at the equator), reported as the number of people per square kilometre ($km^2$). | [32] |
| Gross domestic product (GDP) | A raster with the average gross domestic product (GDP) was downloaded at a resolution of ~1km grid, reported as value in USD. For locations where there was no data, a 1km buffer around the household was created, and the mean value on valid datapoints was extracted and assigned to the household. | [32] |
| Land use and land cover | Data were derived at 10m resolution from the Sentinel-2 Global Land Use/Land Cover (LULC) Timeseries produced by Impact Observatory, Microsoft, and the Environmental Systems Research Institute (ESRI). The global LULC cover map with 11 LULC classes was used to generate six separate rasters for the LULC categories that cover the DR: crops, rangelands, bare ground, trees, flooded vegetation and built/urban area. For each household, a 50m, 100m, 250m, 500m and 1km buffer was generated. The percentage of each LULC raster overlapping with the household buffer was extracted. | [38] |
| River density | The total length (metres) of rivers was extracted from a vector layer of all main rivers in the DR overlapping with the 50m, 100m, 250m, 500m and 1km household buffers. | [32] |
| Flooding-risk area | A vector layer containing a national flooding-risk map, which delimitates areas considered to be at risk of flooding | [a] |

Study procedures and reporting adhered to the STROBE criteria for observational studies.

To assess local clustering of leptospirosis seropositivity at the individual level, we applied spatial scan statistics, which are designed to identify areas with an excess of events compared to the expected distribution of leptospirosis seropositivity assuming no clustering (null hypothesis). By calculating the likelihood of the observed number of events inside and outside a specific area, the spatial scan statistics assess the likelihood that the clustering occurred by chance. The area

with maximum likelihood is defined as the most likely cluster. We analysed our data using the software package FleXScan [44], developed by Tango and Takahashi [43], which uses a flexible-shape spatial scan, allowing for the identification of irregularly shaped clusters. This is particularly useful when considering infection distribution that can follow a non-circular geographic feature (e.g., rivers, catchments, coastal areas) and the irregular boundaries of many geographical areas (i.e., municipalities, districts, provinces).

In this study, as the outcome being investigated (seropositive or seronegative) was binary, we assumed a Bernoulli distribution. The maximum spatial cluster size was set at 100 percent of total population, reflecting the maximum allowed by the software. Additional parameters in FleXScan were the likelihood ratio (LLR) with restrictions and an alpha set at 0.1. Statistically significant clusters were defined as a *p-value* ≤ 0.05. Less stringent *p-values* (≥0.05 and <0.1) were considered marginally statistically significant and used to identify areas with excess risk [44].

FleXScan identifies households inside and outside each cluster, and we used the household location to identify the communities associated with a cluster. We calculated the risk ratio (RR) using the households that were identified as belonging to a cluster and the communities associated with a cluster. For each province, the RR was calculated by comparing i) the proportion of participants who were seropositive from households within the cluster and the proportion of seropositive participants from households outside the cluster ii) the proportion of participants who were seropositive from communities associated with a cluster and the proportion of seropositive participants from communities outside the cluster. The *p*-value was determined using Fisher's exact test.

**Logistic regression analysis.** We employed non-parametric bivariate regression models to compare selected characteristics of participants inside and outside the identified clusters, as the characteristics investigated were not normally distributed. In each model, the dependent variable (*outcome*) was the characteristic being investigated (i.e., distance to major roads, elevation, land use and land cover) and the independent variable (*explanatory variable*) was a binary variable classifying each participant into two groups: residing in household inside or outside the identified cluster. Regression standard errors were clustered at the household level to account for the fact that multiple participants could belong to the same household, potentially leading to correlated outcomes within households. Numerical continuous variables were assessed using linear models. Before conducting the regression models, continuous variables were standardized to have a mean value of zero and a standard deviation of one, preserving their original distribution. All categorical variables with multiple classes were transformed into binary variables and assessed through logistic regression (S1 Table).

A *p-value* ≤0.05 was considered significant. Results are shown as the distribution of each characteristic investigated between participants outside and inside the cluster and the p-value of the bivariate regression.

## Software

We used Esri ArcGIS software v 10.8 (Esri ArcMap 10.8.0.12790. Redlands, CA, USA) [34] to process and extract spatial data, and to create maps showing the results. R version 4.4.0 (2024-04-24 ucrt) [45] was used for data processing and global cluster analysis. FleXScan v3.1.2 (FleXScan v3.1.2: Software for the Flexible Scan Statistic. National Institute of Public Health, Japan, 2013) was used for local cluster analysis. Stata18 software (StataCorp. 2023. Stata Statistical Software: Release 18. College Station, TX: StataCorp LLC) [46] was used for the non-parametric regressions.

## Results

Across the two provinces included in this analysis, there were 2091 participants enrolled in the survey. Thirteen records were excluded because there were incomplete data from the questionnaire, resulting in a final sample of 2,078 participants (802 in Espaillat and 1,276 in SPM). A total of 237 seropositive participants were identified (seroprevalence of 11.4%); 127 (15.8%) in Espaillat and 110 (8.6%) in SPM.

In Espaillat, the four main serogroups were Icterohaemorragiae (58.5%), Australis (21.2%), Canicola (14.1%) and Djasiman (9.7%). In SPM, Icterohaemorrhagiae was the most prevalent serogroup (33.3%) followed by Australis (17.2%) and Pyrogenes, Djasiman and Canicola (all three with 11 cases, 10.0%). Across the two provinces, all but one Pyrogenes cases occurred in SPM (S2 Table).

## Global clustering

In Espaillat and SPM, the semivariograms did not reach a plateau, suggesting a random distribution of seropositive cases of all serogroups combined as well as for the main serogroups individually. S2–S4 Figs present the empirical variograms for all leptospirosis serogroups combined and the main serogroups individually, in each province.

## Local clustering

FleXScan identified one statistically significant cluster in Espaillat and one marginally statistically significant in SPM, for all serogroups combined (Figs 1 and 2). In Espaillat, the identified cluster was located in the southwest of the province and included eight households, with a maximum distance between households of 3.1km (*p-value* 0.044). (Fig 1). The households inside the cluster represent 1.5% of the households surveyed and were home to 7.1% of all seropositive participants in this province. For participants living in a household within the identified cluster, RR was 6.7 (95%CI 5.7-7.9) compared to participants living in a household outside the identified cluster. For participants living in a community where the cluster was identified, the RR was 1.6 (95% CI 1.1-3.2) compared to participants living in a non-cluster community. An additional 11 secondary (non-significant) clusters were identified (S5 Fig).

In SPM, the identified cluster was located in the northeast of the province and included six households, with a maximum distance between households of 10.5km (*p-value* 0.08). (Fig 2) The households inside the cluster represent 0.8% of the households surveyed and were home to 6.4% of all cases in this province. For participants living in a household within the identified cluster, RR was 9.6 (95%CI 5.6-14.2) compared to participants living in households outside the identified cluster. For participants living in a community where the cluster was identified the RR was 2.0 (1.3-2.9) compared to participants living in a non-cluster community. Seven secondary (non-significant) clusters were identified (S6 Fig). Local cluster analysis for main serogroups in both provinces did not identify significant clusters.

## Logistic regression

Results from the non-parametric regression identified that when compared to participants outside the cluster, participants inside the cluster in Espaillat were significantly older (OR 2.22; *p-value* 0.026), worked in outdoor work environments (OR 4.05, *p-value* 0.019), were more likely to live in households with no access to piped water (OR 4.72, *p-value* 0.041), at greater distance to water bodies (OR 1.44, *p-value* 0.023) and health facilities (OR 1.86, *p-value*<0.001 by motorized travel time and OR 1.36, *p-value* 0.001 by walking travel time), and surrounded by greater areas of range land (OR 1.8, *p-value* 0.007) (Table 2). Participants inside the cluster were also less likely to live in households at greater distance to major roads (OR 0.42, *p-value*<0.001), at higher altitudes (OR 0.58, *p-value*<0.001), located in areas of higher population density (OR 0.76, *p-value*<0.001) and higher gross domestic product (GDP) at the community level (OR 0.70, *p-value*<0.001), compared to participants outside the cluster.

Participants inside the cluster in SPM, compared with participants outside the cluster, were significantly more likely to be male (OR 3.89, *p-value* 0.030) and live in households at greater distance to health facilities (OR 4.41, *p-value* 0.044 by motorized travel time and OR 4.67, *p-value* 0.021 by walking travel time) and educational facilities (OR 4.36, *p-value* 0.017), located at higher altitudes (OR 1.15, *p-value* 0.036) and greater soil moisture (OR 1.97, *p-value*<0.001). Participants inside the cluster were also less likely to live in households located in areas of higher population density (OR 0.29, *p-value*<0.001), higher GDP at the community level (OR 0.42, *p-value*<0.001), and surrounded by a greater percentage of built-up areas (OR 0.32, *p-value* 0.019).

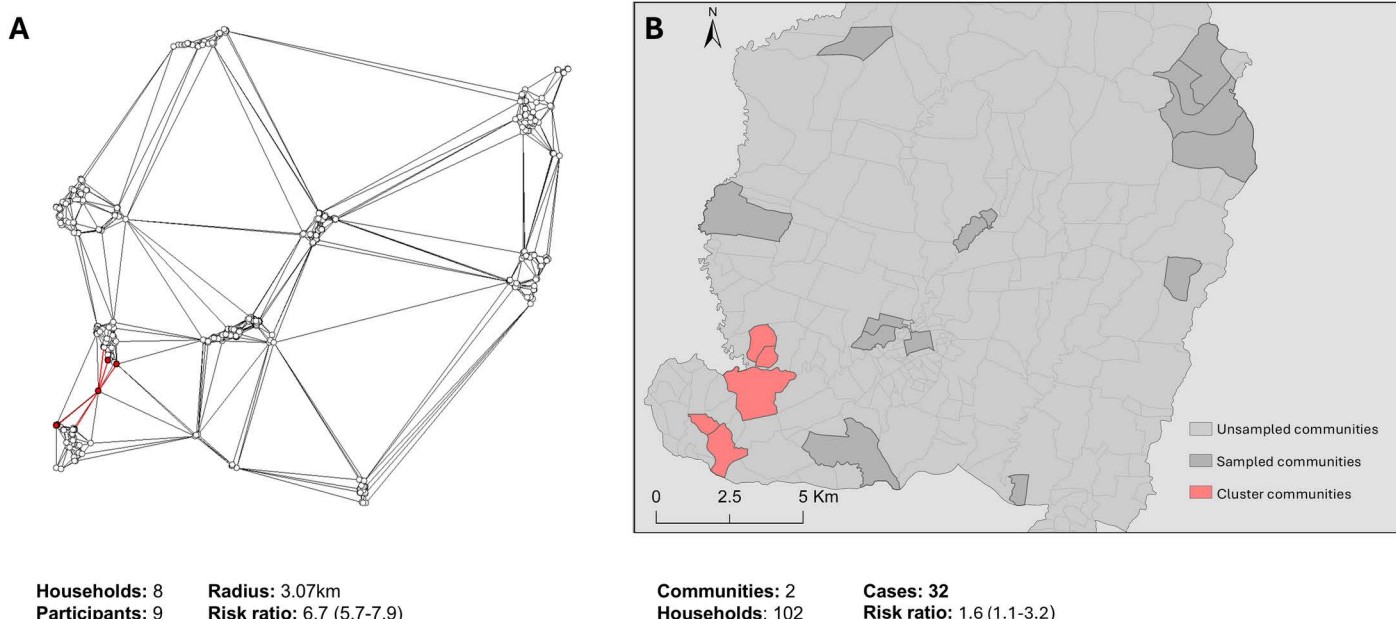

**Households:** 8 **Radius:** 3.07km
**Participants:** 9 **Risk ratio:** 6.7 (5.7-7.9)
**Cases:** 9 *P-value:* *0.0438*

**Communities:** 2 **Cases:** 32
**Households:** 102 **Risk ratio:** 1.6 (1.1-3.2)
**Participants:** 138 *P-value:* 0.014

**Fig 1. Location of clusters identified by flexible spatial scan in Espaillat, Dominican Republic, Jun-Oct 2021.** A) Household's link based on nearest neighbour determined from the Thiessen Polygon, households identified as a hotspot and their connection are highlighted in red. B) Sampled communities in Espaillat with the cluster communities highlighted in red. Base layer from: https://data.humdata.org/dataset/cod-ab-dom.

## Discussion

Our study identified two clusters of leptospirosis seroprevalence in each of the two provinces included in the analysis, with significant associations between cluster and sociodemographic and environmental characteristics. Across both provinces, participants living in households located farther from health facilities and in areas with lower population density and GDP were more likely to belong to the identified clusters. These shared characteristics highlight the role of poverty-associated factors in shaping the epidemiological profile of leptospirosis transmission. However, specific risk factors varied between provinces, reflecting the specific sociodemographic, economic, and ecological contexts. By examining these provincial differences, this study underscores the complexity of leptospirosis transmission dynamics and emphasizes the need for tailored public health interventions to address local drivers of disease.

Characteristics associated with participants living inside the clusters identified by FleXScan can be divided into two groups: common to both provinces and specific to each province. In both provinces, participants living in households located at greater distances from education and health facilities were more likely to be inside the clusters, and participants living in households located in areas of higher population density and higher GDP were less likely to be inside the clusters. These associations suggest that the households inside the identified clusters presented characteristics frequently associated with impoverished communities. The association between leptospirosis and poverty have been extensively studied [2,47–50]. Populations living in resource-poor communities are more prone to overcrowded housing conditions, and have inadequate access to safe water, sanitation, waste management and other infrastructure [7]. These conditions can facilitate the transmission of the *Leptospira* bacteria and offer limited or non-existent barriers to prevent human exposure to contaminated environments [1,7].

In this study, specific risk factors associated with clusters in each province reflected local differences and the complexity of the transmission cycle. Specific characteristics of the population living inside the Espaillat cluster, such as outdoor

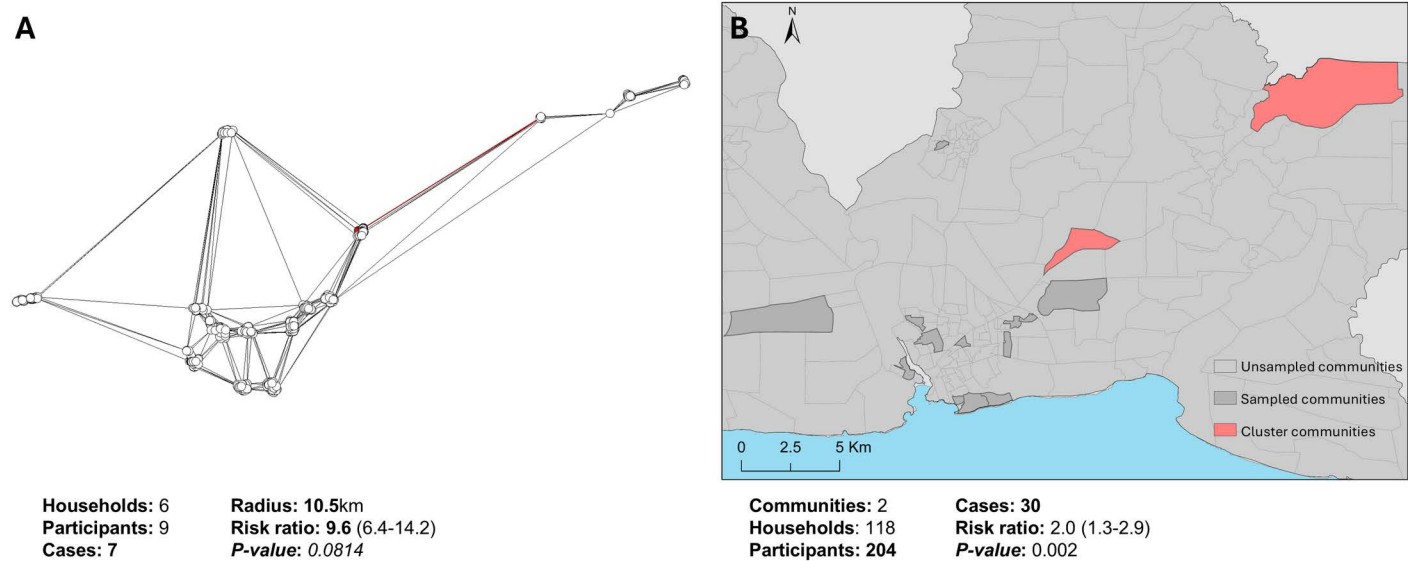

| | | | |
|---|---|---|---|
| **Households:** 6 | **Radius: 10.5**km | **Communities:** 2 | **Cases:** 30 |
| **Participants:** 9 | **Risk ratio: 9.6** (6.4-14.2) | **Households**: 118 | **Risk ratio:** 2.0 (1.3-2.9) |
| **Cases:** 7 | *P-value:* 0.0814 | **Participants:** 204 | *P-value:* 0.002 |

**Fig 2. Location of clusters identified by flexible spatial scan in San Pedro de Macoris, Dominican Republic, Jun-Oct 2021.** A) Household's link based on nearest neighbour determined from the Thiessen Polygon, households identified as a hotspot and their connection are highlighted in red. B) Sampled communities in Espaillat with the cluster communities highlighted in red. Base layer from: https://data.humdata.org/dataset/cod-ab-dom.

work environment, households with limited access to piped water and located at a greater distance to major roads might reinforce the characterisation of impoverished communities [7], but can also suggest a rural setting. Similarly, the association of clusters with a higher percentage of range land surrounding the households indicates a rural setting, implying the presence of livestock as an important reservoirs [5]. In this province, those findings can also mirror a higher percentage of population residing in rural setting [27,51,52]. Although the specific factors associated with the SPM cluster provide fewer factors to characterise the high-risk area, lower percentage of built-up areas surrounding the households and greater distance to the province capital were observed. In SPM, as the population residing in rural settings represents a lower percentage of the total population [27,51,52] and other economic activities have greater importance, these findings might suggest that subsistence farmers, further away from larger cities are more vulnerable to leptospirosis transmission. Finally, clusters in each province presented opposite directions of association with elevation. While Espaillat is located inland, SPM is a coastal province. The SPM cluster is located in the northeast of the province, with greater distance to the coast compared to the other communities included in the study, which may impact the higher median elevation of the cluster. Additionally, the difference in the median elevation between households inside and outside the cluster in Espaillat is small, with a non-negligible overlap of variation range between them. In the two provinces included in this study, although the association was significant, it was not linear and should be interpreted based on the geographical context of each province.

This study has several limitations. The relationship between serogroups and reservoir mammals can be highly specific, and in turn, routes of human exposure are related to exposure to the reservoir mammals. Thus, characteristics of clusters of high leptospirosis seroprevalence are likely to vary based on the predominant serogroup. In our analysis, the small sample size limited local cluster analysis based on the serogroup predominant in each province, which could have provided more comprehensive insights into specific risk factors and drivers of transmission. The main serogroups identified in Espaillat (results shown in the S2 Table), Australis, associated with wild and domestic animals, rats and mice, and Icterohaemorrhagie, mainly associated with rodents [3,53]. In this province, clusters were not associated with self-reported rat exposure, yet several characteristics associated with clusters

**Table 2. Odds ratio of each variable associated with household from a hotspot for all serogroups combined by province, Dominican Republic, Jun-Oct, 2021.**

| | Espaillat | | | | SPM | | | |
|---|---|---|---|---|---|---|---|---|
| | Outside hotspot N = 793 | Inside hotspot N = 9 | OR | *p-value* | Outside hotspot N = 1,267 | Inside hotspot N = 9 | OR | *p-value* |
| Age (years)[1] | 44 (28, 61) | 70 (59, 71) | 2.22 | **0.026** | 34 (21, 52) | 43 (25, 48) | 1.40 | 0.548 |
| Males[2] | 292 (37) | 5 (56) | 2.14 | 0.201 | 430 (34) | 6 (67) | 3.89 | **0.030** |
| Outdoor work environment[2,3] | 187 (24) | 5 (56) | 4.05 | **0.019** | 186 (15) | 3 (33) | 2.91 | 0.155 |
| Primary education or less[2] | 266 (33) | 6 (67) | 3.96 | *0.067* | 380 (30) | 6 (67) | 4.66 | 0.104 |
| Flooding risk area[2] | 326 (41) | 2 (22) | 0.41 | 0.209 | 262 (21) | 0 (0) | NA | NA |
| Freshwater exposure [2] | 9 (1.1) | 0 (0) | NA | NA | 200 (16) | 3 (33) | 2.67 | 0.252 |
| Rats exposure[2] | 4 (0.5) | 0 (0) | NA | NA | 321 (25) | 6 (67) | 5.89 | *0.061* |
| Rural setting[2] | 463 (58) | 4 (44) | 0.57 | 0.445 | 428 (34) | 9 (100) | NA | NA |
| Lack of access to pipped water[2] | 76 (10) | 3 (33) | 4.72 | **0.041** | 310 (24) | 0 (0) | NA | NA |
| Population density (p/km$^2$) [4] | 1,536 (1,822) | 805 (362) | 0.76 | **<0.001** | 3,623 (2,806) | 298 (200) | 0.29 | **<0.001** |
| Distance to major roads (km) [1] | 2.95 (1.88, 5.60) | 2.20 (0.19, 2.26) | 0.42 | **<0.001** | 2.58 (1.15, 5.60) | 6.29 (0.84, 6.30) | 1.75 | 0.381 |
| Distance to waterways (km) [1] | 0.37 (0.17, 0.91) | 0.99 (0.62, 1.02) | 1.44 | **0.023** | 0.38 (0.09, 0.82) | 0.46 (0.38, 0.88) | 1.06 | 0.748 |
| Motorized time travel (min) [1] | 2.8 (1.4, 4.7) | 5.4 (5.4, 7.5) | 1.86 | **<0.001** | 3.4 (1.6, 7.2) | 7.2 (7.2, 18.8) | 4.41 | **0.044** |
| Walking time travel (min) [1] | 26 (11, 48) | 42 (42, 68) | 1.36 | **0.001** | 47 (21, 98) | 98 (98, 232) | 4.67 | **0.021** |
| Dist. to province capital (km) [1] | 0.05 (0.04, 0.08) | 0.05 (0.05, 0.07) | 1.19 | *0.055* | 0.04 (0.02, 0.09) | 0.09 (0.09, 0.19) | 4.19 | **0.008** |
| Dist. to education facility (km)[1] | 0.01 (0.00, 0.02) | 0.01 (0.00, 0.01) | 0.87 | **0.001** | 0.02 (0.01, 0.06) | 0.10 (0.07, 0.16) | 4.36 | **0.017** |
| GDP (1M USD per capita) [1] | 4.6 (0.8, 7.4) | 3.1 (0.8, 3.9) | 0.70 | **<0.001** | 6 (2, 14) | 0 (0, 4) | 0.42 | **<0.001** |
| Elevation (m) [1] | 206 (169, 269) | 191 (159, 219) | 0.58 | **<0.001** | 21 (8, 26) | 57 (20, 57) | 1.15 | **0.036** |
| Precipitation (mm) [1] | 87.0 (85.5, 88.7) | 88.5 (85.7, 88.7) | 1.17 | 0.139 | 81.2 (78.5, 81.2) | 80.4 (80.4, 90.3) | 1.49 | 0.427 |
| Soil moisture (m$^3$.m$^{-3}$) | 0.27 (0.28, 0.28) | 0.28 (0.28, 0.28) | 1.03 | 0.159 | 0.23 (0.23, 0.23) | 0.25 (0.25, 0.26) | 1.97 | **<0.001** |
| Bare ground area (%)[1,5] | 0 (0, 0) | 0 (0, 0) | NA | NA | 0 (0, 0) | 0 (0, 0) | NA | NA |
| Built-up area (%)[1,5] | 61 (39, 94) | 73 (29, 81) | 0.77 | 0.542 | 88 (59, 100) | 55 (3, 65) | 0.32 | **0.019** |
| Cropland (%)[1,5] | 4 (0, 41) | 2 (2, 58) | 1.41 | 0.383 | 0 (0, 9) | 18 (2, 26) | 1.27 | 0.284 |
| Range land (%)[1,5] | 0.0 (0.0, 0.9) | 6.2 (3.7, 8.7) | 1.80 | **0.007** | 0.0 (0.0, 5.0) | 17 (11, 17) | 15.03 | *0.081* |
| NDVI[4] | 0.77 (0.05) | 0.76 (0.00) | 0.61 | **0.002** | 0.80 (0.04) | 0.81 (0.03) | 1.15 | *0.062* |

[1]Median (IQR);

[2]n (%).

[3]Outdoor work environment included active workers with exclusively outdoor work environment and mixed indoor and outdoor work environment.

[4]Mean (SD);

[5]Calculated on a 250m buffer around each household. *NA*: results not available due to sample size.

were indicative of remote impoverished, agricultural communities, which could be associated with the presence of reservoirs of Australis and Icterohaemorrhagie. In SPM, the main serovar identified was Australis, followed by Canicola, Djasiman, Icterohaemorrhagie and Pyrogenes equally distributed, which suggests a broader range of reservoirs driving transmission in the province. Our data were obtained from a cross-sectional survey, although this study design has the advantage of providing a more extensive population-level seroprevalence characterisation, it was limited to one point in time. Water-borne diseases such as leptospirosis are strongly affected by climatic conditions, and transmission might vary according to seasonal variations in rainfall and temperature [1,9]. However, our study did not identify association between clusters and precipitation. The survey design may have impacted the semivariogram results. The sampling design aimed for a national representative and widespread spatial distribution of sampled communities as described previously [29]. However, by sampling few communities in each province,

few households in each community, and including all members of each household sampled, the spatial distribution of the final sample was, to some extent, clustered. To identify positive spatial autocorrelation, the seroprevalence clustering needed to overcome the clustered spatial distribution of our sample, which might have been limited due the reduced sample size in each province. A clustered spatial pattern of leptospirosis cases has been reported on several occasions, by both global and local clustering methods [17–21,54].

This study focused on leptospirosis seroprevalence in Espaillat and SPM, yet the concepts and methods could be applied to other locations in the country and the other regions, as well as to other infectious diseases strongly driven by environmental and sociodemographic factors in these provinces. Our results reinforce the findings of previous studies that identified clustered spatial pattern of leptospirosis and provide empirical evidence that drivers of transmission can be highly specific to the context of each province. In both provinces, the identified clusters of leptospirosis seroprevalence were associated with characteristics which suggests higher transmission in impoverished communities and most likely in rural settings. However, the translation of this insight into modifiable factors varied between provinces. Aiming to achieve cost-effective prevention and control measures will require public health and environmental interventions specific to each context. Further studies exploring risk factors and drivers associated with predominant serogroups could help enhance future targeted interventions.

## Supporting information

**S1 Text. Supporting information on study design and participant selection.**
(DOCX)

**S1 Table. Adjustments conducted to transform multiple class categorical variables into binary variables.**
(XLSX)

**S2 Table. Proportion of positive Leptospira microscopic agglutination tests by main primary reacting serogroup in Espaillat and San Pedro de Macoris Provinces, Dominican Republic, Jun-Oct 2021.**
(XLSX)

**S1 Fig. Parameters of the semivariogram.** Data from a lymphatic filariasis serosurvey previously published by Lau et al, 2014 [55], was used to illustrate the semivariogram parameters.
(TIF)

**S2 Fig. Semivariogram of the participants seropositive for all leptospirosis serogroup combined by province, Dominican Republic, Jun-Oct 2021.** A. Espaillat, B. San Pedro de Macoris.
(TIF)

**S3 Fig. Semivariogram of the leptospirosis seropositive participants in Espaillat, by main serogroup, Dominican Republic, Jun-Oct 2021.** Serogroups A. Icterohaemorrhagiae B. Djasiman C. Canicola D. and Australis.
(TIF)

**S4 Fig. Semivariogram of the leptospirosis seropositive participants in San Pedro de Macoris, by main serogroup, Dominican Republic, Jun-Oct 2021.** Serogroups A. Icterohaemorrhagiae B. Djasiman C. Canicola D. and Australis.
(TIF)

**S5 Fig. Location of clusters (primary in red and secondary in orange) identified by flexible spatial scan (FleXScan) in Espaillat, Dominican Republic, Jun-Oct 2021.** Base layer from: https://data.humdata.org/dataset/cod-ab-dom.
(TIF)

**S6 Fig. Location of clusters (primary in red and secondary in orange) identified by flexible spatial scan (FleXScan) in San Pedro de Macoris, Dominican Republic, Jun-Oct 2021.** Base layer from: https://data.humdata.org/dataset/cod-ab-dom.
(TIF)

## Acknowledgments

We would like to thank the many study participants who volunteered to participate in this study. We would also like to thank the study staff who collected the field data, the Dominican Republic Ministry of Health and Social Assistance, and the Pedro Henriquez Ureña National University, for their commitment and support for the study. Finally, we would like to thank Dr Gregorio Antonio Rosario Michel and the valuable team working in the Servicio Geologico Nacional for providing the flooding-risk map.

## Author contributions

**Conceptualization:** Helen J. Mayfield, Angela M. Cadavid Restrepo, Ronald Skewes-Ramm, Eric J. Nilles, Colleen L Lau.

**Data curation:** Beatris Mario Martin, Cecilia J. Then Paulino, Marie Caroline Etienne, Michael de St. Aubin, Devan Dumas, Salomé Garnier, William Duke, Farah Peña, Gabriela Abdalla, Lucia de la Cruz, Bernarda Henríquez, Margaret Baldwin, Adam Kucharski.

**Formal analysis:** Beatris Mario Martin, Benn Sartorius.

**Funding acquisition:** Eric J. Nilles, Colleen L Lau.

**Investigation:** Beatris Mario Martin, Benn Sartorius, Helen J. Mayfield, Angela M. Cadavid Restrepo, Eric J. Nilles, Colleen L Lau.

**Methodology:** Beatris Mario Martin, Benn Sartorius, Colleen L Lau.

**Project administration:** Beatris Mario Martin, Benn Sartorius, Colleen L Lau.

**Resources:** Cecilia J. Then Paulino, Ronald Skewes-Ramm, Farah Peña, Eric J. Nilles, Colleen L Lau.

**Supervision:** Benn Sartorius, Eric J. Nilles, Colleen L Lau.

**Validation:** Benn Sartorius, Angela M. Cadavid Restrepo, Eric J. Nilles, Colleen L Lau.

**Visualization:** Beatris Mario Martin, Benn Sartorius, Colleen L Lau.

**Writing – original draft:** Beatris Mario Martin.

**Writing – review & editing:** Beatris Mario Martin, Benn Sartorius, Helen J. Mayfield, Angela M. Cadavid Restrepo, Behzad Kiani, Cecilia J. Then Paulino, Marie Caroline Etienne, Michael de St. Aubin, Devan Dumas, Salomé Garnier, William Duke, Farah Peña, Gabriela Abdalla, Lucia de la Cruz, Bernarda Henríquez, Margaret Baldwin, Adam Kucharski, Eric J. Nilles, Colleen L Lau.

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
