## [Decision Letter · Decision Letter 0]

19 Mar 2025

Uncovering Leptospirosis Clusters and associated risk factors in the Dominican Republic through Geospatial Analysis

Dear Dr. Mario Martin,

Thank you for submitting your manuscript to PLOS Neglected Tropical Diseases. After careful consideration, we feel that it has merit but does not fully meet PLOS Neglected Tropical Diseases's publication criteria as it currently stands. Therefore, we invite you to submit a revised version of the manuscript that addresses the points raised during the review process.

Please submit your revised manuscript within 60 days May 18 2025 11:59PM. If you will need more time than this to complete your revisions, please reply to this message or contact the journal office at plosntds@plos.org. Please include the following items when submitting your revised manuscript:

We look forward to receiving your revised manuscript.

Kind regards,

Joseph M. Vinetz

Section Editor

Joseph Vinetz

Section Editor

Shaden Kamhawi

co-Editor-in-Chief

Paul Brindley

co-Editor-in-Chief

**Journal Requirements:**

At this stage, the following Authors/Authors require contributions: Beatris Mario Martin, Benn Sartorius, Helen J. Mayfield, Angela M. Cadavid Restrepo, Behzad Kiani, Cecilia J. Then Paulino, Marie Caroline Etienne, Ronald Skewes-Ramm, Michael de St. Aubin, Devan Dumas, Salome Garnier, William Duke, Farah Peña, Gabriela Abdalla, Lucia de la Cruz, Bernarda Henríquez, Margaret Baldwin, Adam Kucharski, Eric J. Nilles, and Colleen L Lau. Please ensure that the full contributions of each author are acknowledged in the "Add/Edit/Remove Authors" section of our submission form.

- ® on page: 12.

4) Please amend your detailed Financial Disclosure statement. This is published with the article. It must therefore be completed in full sentences and contain the exact wording you wish to be published. State what role the funders took in the study. If the funders had no role in your study, please state: "The funders had no role in study design, data collection and analysis, decision to publish, or preparation of the manuscript.".

**Reviewers' Comments:**

Reviewer's Responses to Questions

**Key Review Criteria Required for Acceptance?**

**Methods:**

-Are the objectives of the study clearly articulated with a clear testable hypothesis stated?

-Is the study design appropriate to address the stated objectives?

-Is the population clearly described and appropriate for the hypothesis being tested?

-Is the sample size sufficient to ensure adequate power to address the hypothesis being tested?

-Were correct statistical analysis used to support conclusions?

-Are there concerns about ethical or regulatory requirements being met?

Reviewer #1: The study focuses on Espaillat and SPM but generalised the data to represent leptospirosis clusters for the entire country of Dominican Republic. Would it be possible to include data from other areas of the country? As you rightly concluded,the drivers of leptospirosis transmission can be highly specific to the context of each province and as such, the characteristics/risk factors may very likely be different if other areas in the Dominican Republic were to be included in this study. Otherwise, I suggest revising the title to reflect the presented data.

**Results:**

-Does the analysis presented match the analysis plan?

-Are the results clearly and completely presented?

-Are the figures (Tables, Images) of sufficient quality for clarity?

Reviewer #1: Were the participants healthy at the point of blood collection? From the writing, it is clear that this study was conceived as a result of the studies in Refs. 27 and 30, or maybe the other way around. Nonetheless, it would be helpful to the readers if there could be some form of linkage from the published papers to the present paper, so that we can understand the study design and not necessarily having to refer to the published papers.

**Conclusions:**

-Are the conclusions supported by the data presented?

-Are the limitations of analysis clearly described?

-Do the authors discuss how these data can be helpful to advance our understanding of the topic under study?

-Is public health relevance addressed?

Reviewer #1: The main results described MAT findings of the seropositive participants in SPM and Espaillat. Additionally, the authors mentioned that the relationship between serogroups and reservoir mammals can be highly specific, and in turn, routes of human exposure are related to exposure to the reservoir mammals. I would expect the authors to discuss on the possible reservoir animals present in the hot spots or what could be other possible explanations for those serogroups to be present in the hot spots.

**Editorial and Data Presentation Modifications?**

Reviewer #1: (No Response)

**Summary and General Comments:**

Reviewer #1: The study presents insights into leptospirosis transmission in the Dominican Republic and employs a robust methodology. However, there are areas where the study could be improved.

PLOS authors have the option to publish the peer review history of their article (what does this mean? ). If published, this will include your full peer review and any attached files.

**Do you want your identity to be public for this peer review?** For information about this choice, including consent withdrawal, please see our Privacy Policy .

Reviewer #1: No

**Figure resubmission:**

**Reproducibility:**



---

## [Editor Report · Decision Letter 1]

1 May 2025

Dear Dr Mario Martin,

We are pleased to inform you that your manuscript 'Geospatial analysis of leptospirosis and risk factors in two provinces of the Dominican Republic' has been provisionally accepted for publication in PLOS Neglected Tropical Diseases.

Best regards,

Joseph M. Vinetz

Section Editor

Joseph Vinetz

Section Editor

Shaden Kamhawi

co-Editor-in-Chief

Paul Brindley

co-Editor-in-Chief

---

## [Editor Report · Acceptance letter]

Dear Dr Mario Martin,

We are delighted to inform you that your manuscript, "Geospatial analysis of leptospirosis clusters and risk factors in two provinces of the Dominican Republic," has been formally accepted for publication in PLOS Neglected Tropical Diseases.

Best regards,

Shaden Kamhawi

co-Editor-in-Chief

Paul Brindley

co-Editor-in-Chief
